# A bacteriocin expression platform for targeting pathogenic bacterial species

Jack W. Rutter [1,4], Linda Dekker [1,4], Chania Clare [1], Zoe F. Slendebroek[1], Kimberley A. Owen [1], Julie A. K. McDonald[2], Sean P. Nair [3], Alex J. H. Fedorec [1] & Chris P. Barnes [1] ✉

Bacteriocins are antimicrobial peptides that are naturally produced by many bacteria. They hold great potential in the fight against antibiotic resistant bacteria, including ESKAPE pathogens. Engineered live biotherapeutic products (eLBPs) that secrete bacteriocins can be created to deliver targeted bacteriocin production. Here we develop a modular bacteriocin secretion platform that can be used to express and secrete multiple bacteriocins from non-pathogenic *Escherichia coli* host strains. As a proof of concept we create Enterocin A (EntA) and Enterocin B (EntB) secreting strains that show strong antimicrobial activity against *Enterococcus faecalis* and *Enterococcus faecium* in vitro, and characterise this activity in both solid culture and liquid co-culture. We then develop a Lotka-Volterra model that can be used to capture the interactions of these competitor strains. We show that simultaneous exposure to EntA and EntB can delay *Enterococcus* growth. Our system has the potential to be used as an eLBP to secrete additional bacteriocins for the targeted killing of pathogenic bacteria.

The microbiota has a profound impact on human health and these communities are implicated in many pathological states. This impact has created growing interest in methods that can manipulate the human host-microbiota system to combat disease[1–4]. One approach is to use engineering biology techniques to construct engineered live biotherapeutic products (eLBPs)[5]. These eLBPs are typically, although not exclusively, bacterial strains that have been modified to perform a therapeutic purpose, e.g. the production of a therapeutic molecule. Currently many eLBPs are undergoing clinical trials to assess their efficacy in human patients, with indications for cancer and hyperoxaluria, amongst others[6]. Another promising application of eLBPs is the production of antimicrobial peptides (AMPs) that can be used to target antibiotic-resistant, pathogenic species.

Alongside technologies such as phage therapy[7], AMPs have emerged as a potential alternative to traditional antibiotics in the face of concerns over antibiotic resistance[8]. AMPs are a class of small peptides that display antimicrobial activity against a range of bacteria,

fungi, viruses and parasites. Within this study we focus on a subset of AMPs known as bacteriocins[9]. All major lineages of bacteria are thought to produce at least one bacteriocin[10] for self-preservation or to produce a competitive advantage in polymicrobial environments[11]. Bacteriocins are ribosomally synthesised and generally show potent activity against a narrow spectrum of bacteria closely related to the producing species (although some broad-spectrum bacteriocins do exist)[12]. This narrow spectrum of activity is a desirable trait for human health applications; as narrow spectrum treatments allow for the targeted removal of pathogenic species, while limiting the disruptive impact the antimicrobial has on the native microbiota.

To date, bacteriocins have been used in a range of biotechnological and industrial applications[13]. They may also be used to target antibiotic-resistant pathogenic bacteria[14,15], via secretion from engineered strains. However, a suitable secretion system is required to ensure that the expressed bacteriocin is able to exit the host cell and target pathogenic bacteria in the environment. It has previously been

[1]Department of Cell and Developmental Biology, University College London, London, UK. [2]Centre for Bacterial Resistance Biology, Department of Life Sciences, Imperial College London, London, UK. [3]Department of Microbial Diseases, UCL Eastman Dental Institute, University College London, London, UK. [4]These authors contributed equally: Jack W. Rutter, Linda Dekker. ✉e-mail: christopher.barnes@ucl.ac.uk

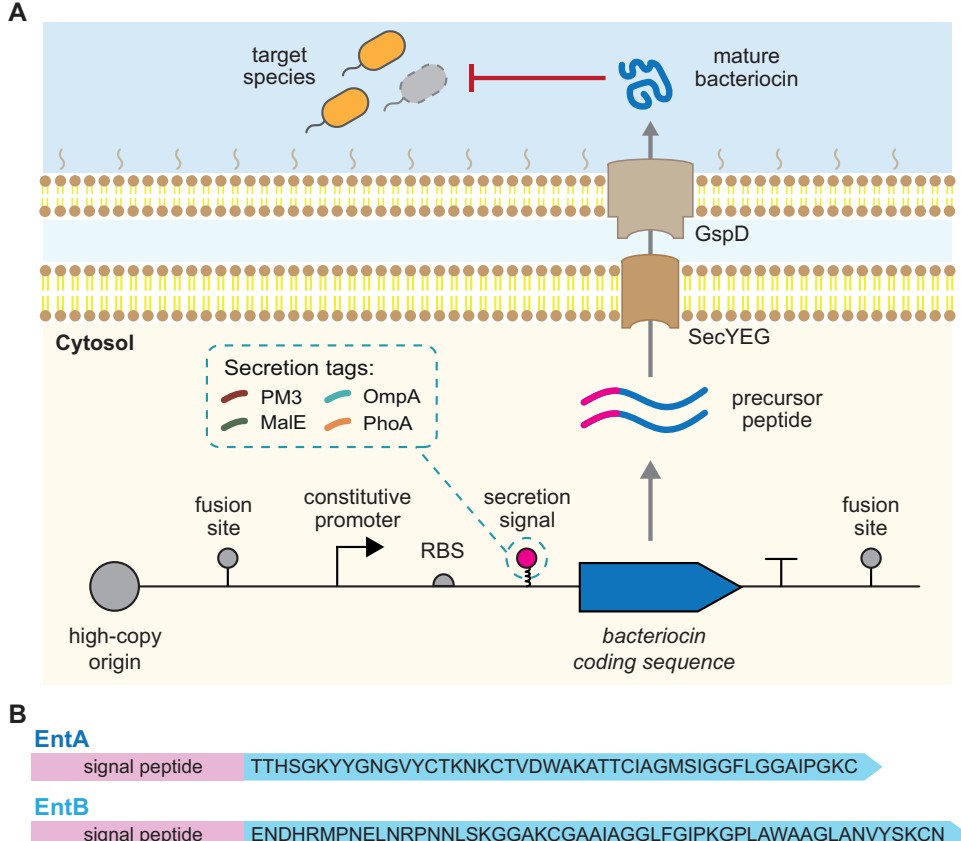

**Fig. 1 | Overview of the bacteriocin expression platform. A** A high-copy plasmid codes for constitutive expression of a specific bacteriocin fused with one of four different secretion signals. The expressed bacteriocins are secreted from the host cells and can target susceptible species in the extracellular environment. The secretion signals used in this study have previously been reported to exit the cell via the SecYEG inner-membrane and the GspD outer-membrane pore. **B** The amino acid sequences of the EntA and EntB bacteriocins used in this study.

shown that secretion signals affect the synthesis and secretion of proteins in an unpredictable manner[16]. Although some studies have applied machine learning techniques to the classification and de novo design of secretion peptides[17,18], it is still a non-trivial task to predict the optimal secretion signal for a specific peptide sequence[19].

Bacteriocin producing systems have been previously reported[20–22]. Geldart et al. designed the pMPES2 platform, which repurposed the microcin V (MccV) secretion system to express multiple bacteriocins from a single operon under the control of the constitutive ProTeOn+ promoter[23,24]. The authors showed that simultaneously producing multiple bacteriocins was able to slow the development of resistance in the target species. The strain also reduced *Enterococcus faecium* 8E9 and *Enterococcus faecalis* V583R colonisation in a mouse model of vancomycin-resistant enterococci (VRE) infection. Although the system was successful, it was shown that the operon layout of multiple-bacteriocin-producing strains greatly affected the antimicrobial activity of their pMPES2 platform[24].

In this study we report the development of a new bacteriocin secretion platform that can be used to target pathogenic species. As a proof of concept we target two *Enterococcus* species as pathogens of interest in vitro, *E. faecalis* and *E. faecium*. One of the most prevalent enterococci strains found in the human gut[25], *E. faecalis* is a Gram-positive, opportunistic pathogen that is associated with endocarditis, systemic and urinary tract infections[10]. Furthermore, there is some evidence that *E. faecalis* is involved in the development of colorectal cancer[25], although there is strong debate whether it plays a harmful or protective role[26–30]. *E. faecium*, identified as an ESKAPE pathogen by the World Health Organisation, has been shown to exhibit resistance to a range of antibiotics and is the most common causative agent of VRE

infections[31–33]. We build on previous works in two main ways. We develop a modular platform based on the CIDAR MoClo assembly standard[34], which allows for flexibility in part interchange. Additionally we explore four different secretion signal peptides, MalE, OmpA, PhoA and PM3 (a modified version of the PelB secretion tag[35]), to add another tunable parameter for eLBP delivery. We then use co-culture assays and Lotka-Volterra modelling to gain insights into the growth dynamics of these strains. Within this study we characterise our platform extensively in vitro, it should be noted that further work will be needed to confirm the activity of our eLBPs under in vivo conditions.

## Results

### Development of the AMP system

We started by designing a modular platform for the expression and secretion of bacteriocins. A summary of this bacteriocin expression platform is given in Fig. 1A. Expressed bacteriocins are exported from the host strain using a secretion tag and can then target pathogenic strains in the extracellular environment. Four secretion tags were tested to determine if the level of antimicrobial activity differed between each tag. The bacteriocin coding sequence linked to the secretion tag is under the control of a constitutive promoter.

Enterocin A (EntA) and Enterocin B (EntB) were chosen as candidate bacteriocins to show the efficacy of our AMP system. The peptide sequences for both are given in Figure 1B. The antimicrobial activity of these bacteriocins against *E. faecalis* was confirmed via chemically synthesised peptides (Supplementary Fig. S1). We also confirmed that EntA and EntB did not show antimicrobial activity against *E. coli* NEB®express cells, when exposed to 4 µg of synthetic bacteriocin (Supplementary Fig. S1C). Growth curves of *E. faecalis* cultures grown

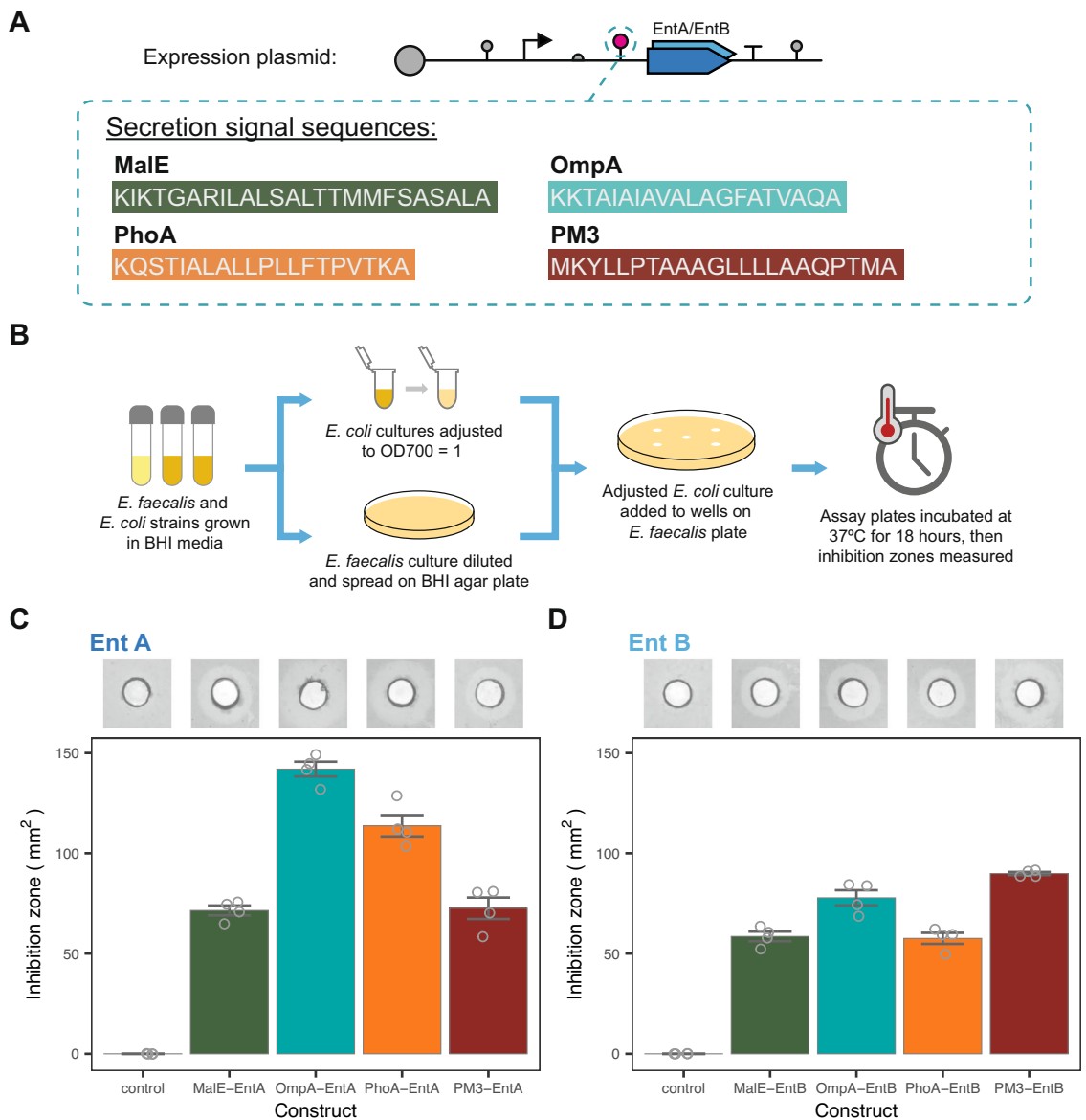

**Fig. 2 | Inhibition zone characterisation of the individual bacteriocin-expressing strains. A** The amino acid sequences of the four secretion signals screened in this study. **B** The experimental protocol used for solid culture inhibition assays. The measured inhibition zones for EntA **C** and EntB **D** secreting strains with all four secretion tags. The control indicates the host strain with no bacteriocin expression plasmid. Insets show representative images of the inhibition zones (central area indicates loading well). Bars indicate mean values ± SE ($n = 4$ biological repeats). Source data are provided as a source data file.

in the presence of EntA, EntB or a mixture of both (EntAB) showed that EntAB was able to delay the onset of *E. faecalis* growth at higher concentrations (Supplementary Fig. S1D).

A live/dead assay in which *E. faecalis* cells were exposed to synthetic EntA or EntB was performed to confirm bactericidal activity (Supplementary Fig. S2). EntA treated cells did not have a higher percentage of dead cells than the negative control, whilst EntB treated samples had a significantly higher percentage of dead cells than the control.

### Screening of bacteriocin secretion constructs

We tested the efficacy of our platform against *E. faecalis* in both solid and liquid culture assays. The sequences of the four secretion signals are given in Figure 2A. The predicted cleavage site of these secretion tags was confirmed using the online SignalP 6.0 tool (Supplementary Fig. S3)[36]. An inhibition assay was used to confirm the antimicrobial activity of our engineered strains against *E. faecalis*. All of the engineered strains were found to produce a zone of inhibition. The largest

inhibition zone was found with the OmpA-EntA strain and the smallest with the PhoA-EntB strain. No inhibition zone was observed for the control construct, which did not contain a bacteriocin expression plasmid (Fig. 2C, D). Inhibition from the expected proteins was also confirmed by a Tricine SDS-PAGE activity gel (Supplementary Fig. S4).

Competition assays between our engineered *E. coli* and *E. faecalis* were performed in liquid culture (Fig. 3A). Co-culture dynamics were extracted from bulk measurements using a fluorescent label in the *E. coli* strain (Supplementary Fig. S5)[37]. The competitive exclusion of *E. coli* by *E. faecalis*, seen with the control, is counteracted, with varying degrees of efficacy, by the secretion of bacteriocins (Fig. 3B). In order to rank the performance of the strains the proportion of *E. faecalis* at 10 hours was chosen as a summary statistic (Fig. 3C). All bacteriocin secreting strains suppressed the growth of *E. faecalis* in comparison with the control strain. In addition, all EntA secreting strains performed better than their EntB counterparts, agreeing with the activity of the synthesised bacteriocins (Supplementary Fig. S1) and previous studies that suggest EntA has greater antimicrobial activity[24].

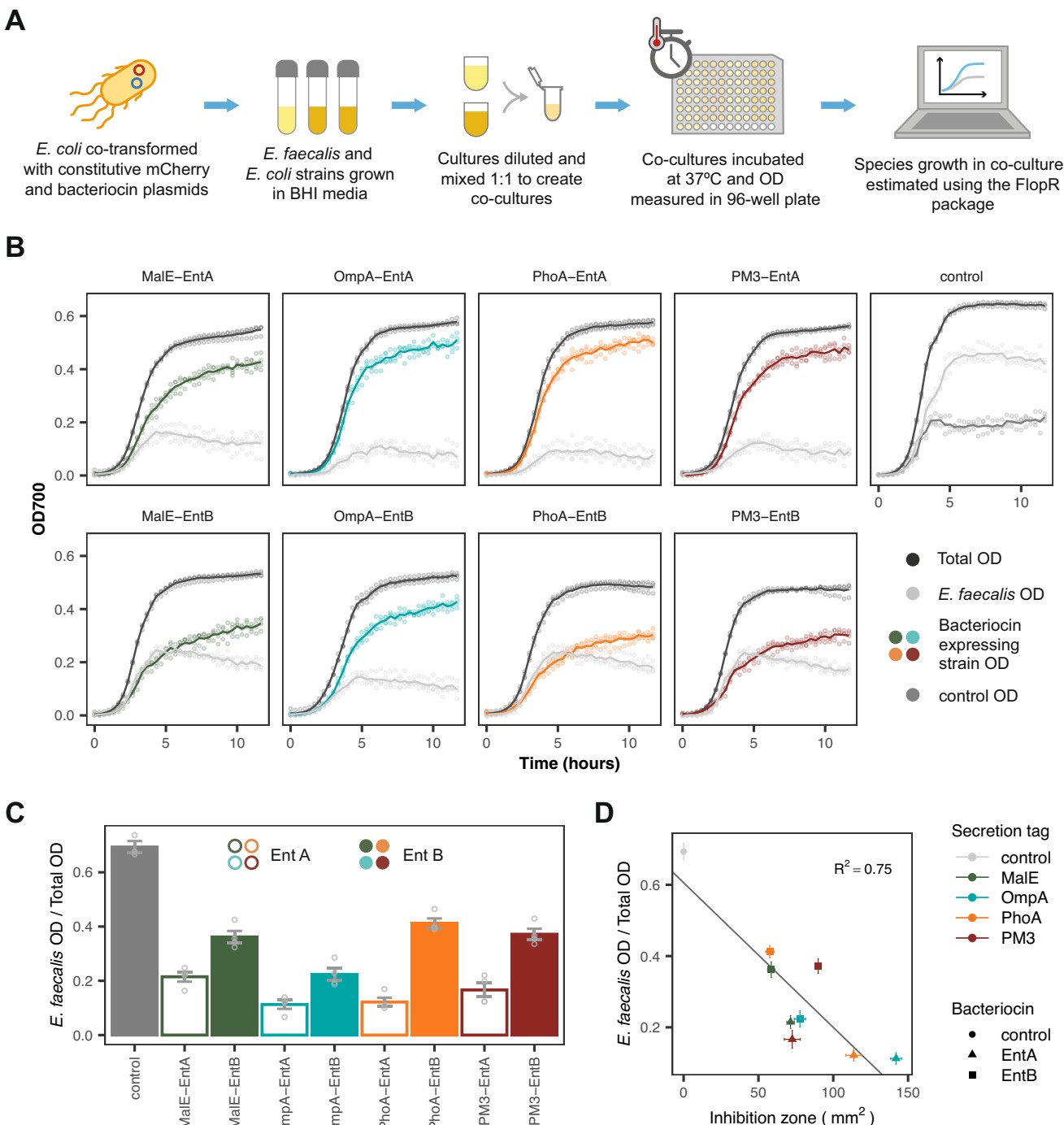

**Fig. 3 | Liquid co-culture characterisation of the individual bacteriocin expressing strains. A** The experimental protocol used for co-culture growth assays. **B** Growth curves of the competitor strains in co-culture. The labels indicate the bacteriocin construct produced by the engineered competitor strain. Lines indicate mean and points show individual measurements (*n* = 4 biological repeats). **C** The estimated ratio of *E. faecalis* OD over total OD at 10 hours, when grown in co-culture with the indicated strain. Bars indicate mean values ± SE (*n* = 4 biological repeats). **D** Comparison of the solid culture inhibition zone vs the 10 hour ratio for each of the co-culture conditions. Solid line shows linear regression fit, with the $R^2$ value labelled. Points give mean value ± SE (*n* = 4 biological repeats). Source data are provided as a source data file.

A comparison of the inhibition zones and co-culture ratio showed that the size of the inhibition zone for each strain was not a good indicator of the growth seen in co-culture (Fig. 3D).

We further investigated the effects of initial culture density and *E. faecalis*: *E. coli* (Ef:Ec) seeding ratios using the PM3-EntA secreting strain. Decreasing the initial culture density and increasing the Ef:Ec seeding ratio resulted in higher levels of *E. faecalis* growth

(Supplementary Fig. S6C). We hypothesise this may be due to the need to reach a threshold level of bacteriocin that can overcome competitive exclusion, which favours the faster growing *E. faecalis* strain. Also, the *E. faecalis* proportion remained relatively stable from six to ten hours (Supplementary Fig. S6A, B); suggesting that for the conditions tested, final co-culture composition is determined before the co-culture reaches the stationary phase of growth.

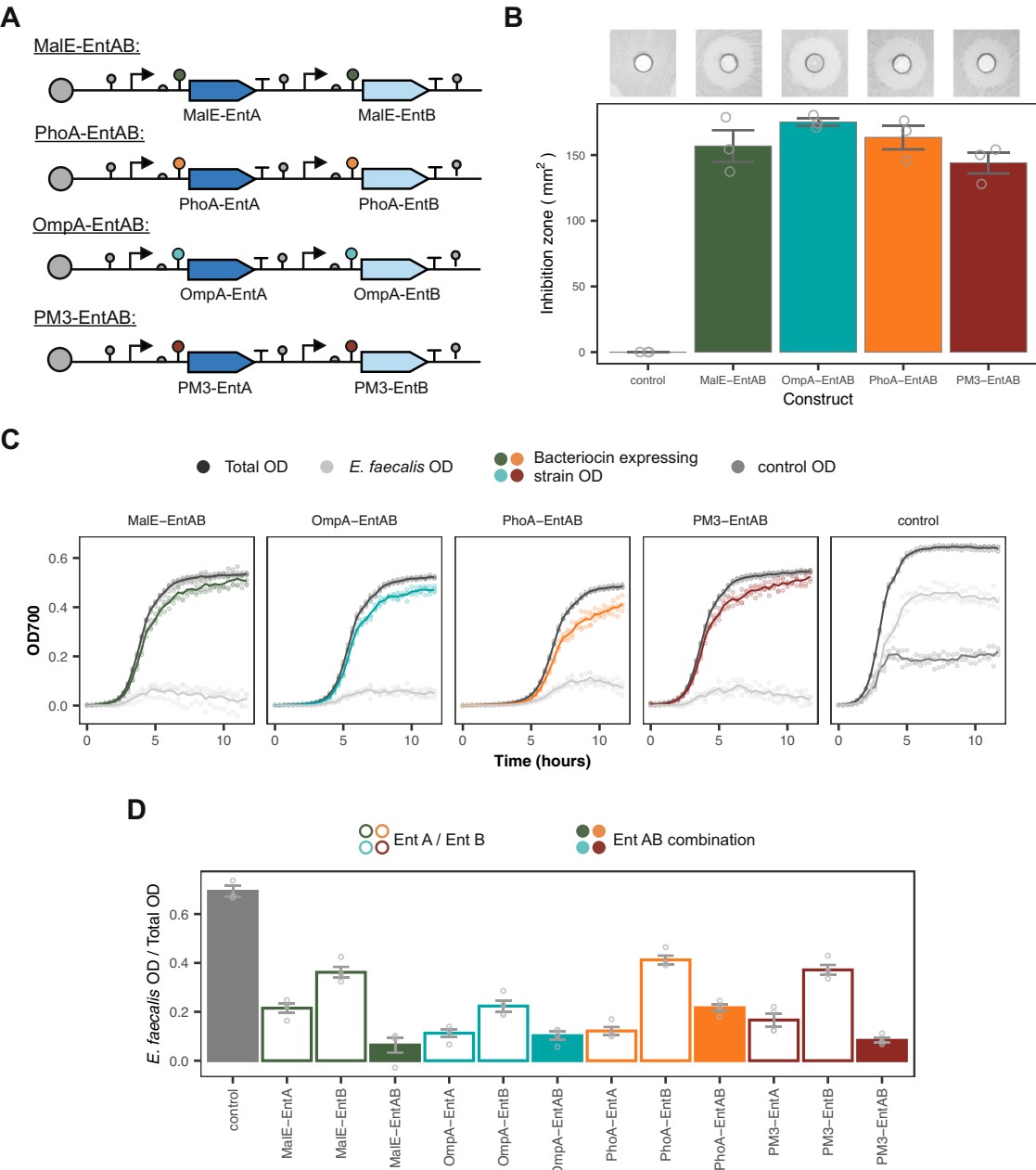

**Fig. 4 | Full characterisation of the dual-bacteriocin expressing strains.**
**A** Plasmid layout of the co-expressing strains. **B** Inhibition zones of the dual-secreting strains. Insets show representative images of the inhibition zones. Bars indicate mean values ± SE ($n = 3$ biological repeats). **C** Growth curves of strains in co-culture. Labels give bacteriocin construct produced by competitor strain. Lines indicate mean and points show individual measurements ($n = 4$ biological repeats). **D** Estimated ratio of *E. faecalis* OD over total OD at 10 h for the dual-bacteriocin strains compared to the ratios given in Figure 3C, when grown in co-culture. Bars give mean values ± SE ($n = 4$ biological repeats). Source data are provided as a source data file.

## Impact of dual-bacteriocin secretion

Next, we set out to explore the effect of secreting multiple bacteriocins simultaneously, as we hypothesised that this would lead to an increase in antimicrobial activity. Using our plasmid system we constructed four EntAB co-expressing strains that used the same secretion tags for each bacteriocin (Fig. 4A). The co-culture assays, once again, demonstrate that the EntAB-expressing strains are able to overcome competitive exclusion and suppress *E. faecalis* growth (Fig. 4C). The proportion of *E. faecalis* in the co-cultures at 10 hours is less in the MalE-EntAB and PM3-EntAB dual-bacteriocin assays compared to the single bacteriocin assays (Fig. 4D), demonstrating that expression of both bacteriocins improves the activity of these strains.

To check that bacteriocin production did not significantly increase the metabolic burden on the host strain, the growth curves of each strain in monoculture (Supplementary Fig. S7A) were fit with a Gompertz model (Supplementary Fig. S7B). As expected, the control showed the fastest growth rate of the engineered *E. coli* strains. The control contains only a single fluorescent plasmid and therefore does not have to divert cellular resources for bacteriocin production. Of the bacteriocin-producing strains, OmpA-EntAB and PhoA-EntAB were found to have the slowest growth rates and longest lag times (Supplementary Fig. S7C, E, respectively).

To show the wider application of our system we screened the PM3-EntA expression plasmid in three additional *E. coli* host strains (BW25113, NEB®5-$\alpha$ and Nissle 1917). *E. faecalis* inhibition zones were

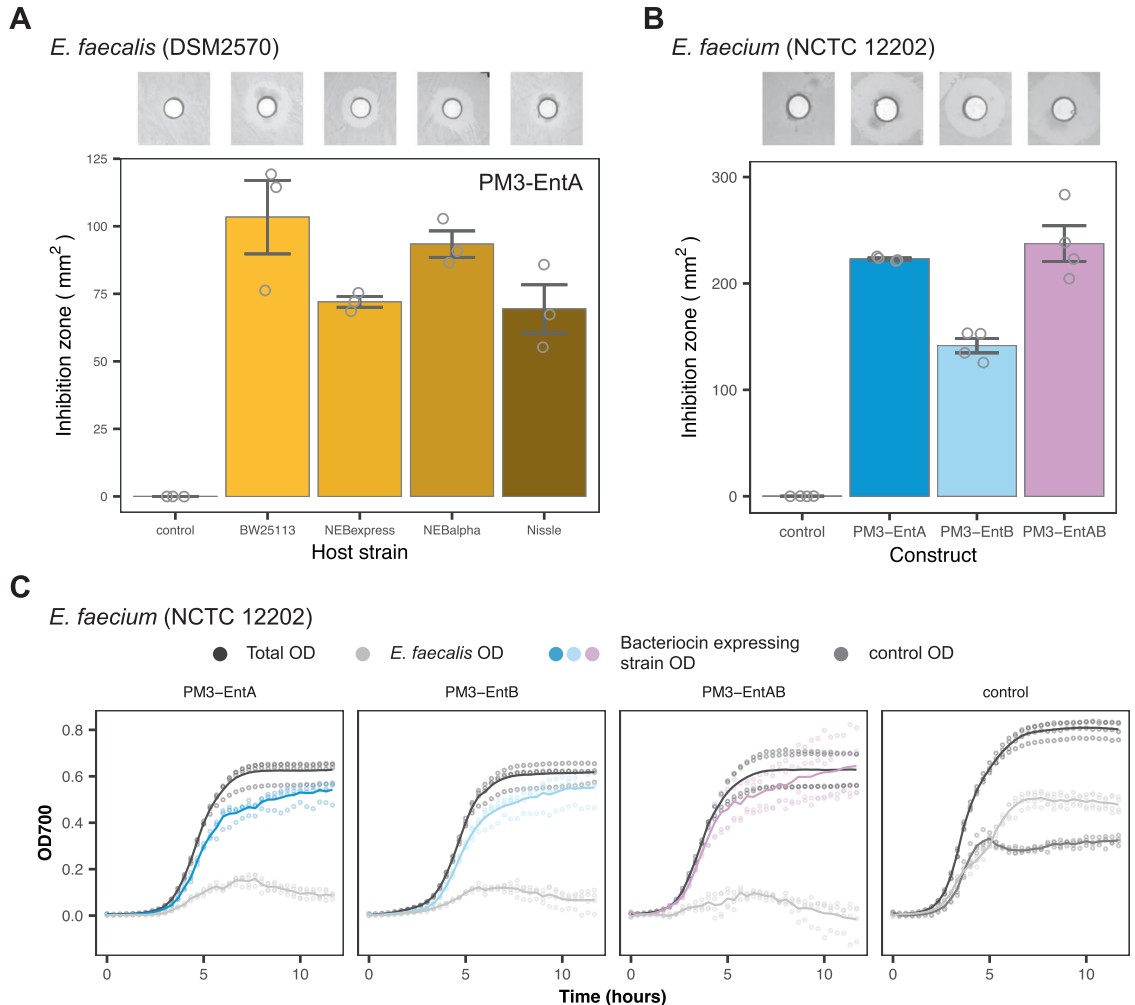

**Fig. 5 | Wider application of the expression system, in alternative hosts and targeting vancomycin-resistant *E. faecium*. A** Inhibition zone assay of the PM3-EntA construct in multiple *E. coli* host strains against *E. faecalis*. Insets show representative images of the inhibition zones observed. Bars give mean values ± SE (*n* = 3 biological repeats). **B** Inhibition zone assays for the given constructs against a Vancomycin-resistant *E. faecium* isolate. Bars give mean values ± SE (*n* = 4 biological repeats). **C** The growth curves of *E. faecium* grown in co-culture with the labelled competitor strain. Lines indicate mean and points show individual measurements (*n* = 4 biological repeats). Source data are provided as a source data file.

seen at similar levels for all host strains (Figure 5A). In addition, our system showed antimicrobial activity against a VRE isolate of *E. faecium* (Fig. 5B, C)[33]. As with other antimicrobials, bacteria can develop resistance to bacteriocins over time. We used a 12 h passage experiment to test for *E. faecalis* growth over an extended time period. For all PM3- strains, we found that *E. faecalis* was able to out compete the engineered strains after 48 h in co-culture (Supplementary Fig. S8).

**Modelling two-strain co-culture dynamics**

We used mechanistic modelling and Bayesian statistics in order to gain further insights into the dynamics of the bacteriocin interaction. We jointly fitted the data from the PM3 system including monocultures of *E. faecalis* and *E. coli* single and dual bacteriocin production strains in co-culture with *E. faecalis* (extended SI Methods). Five different models were developed in total. Three of the models were based on a linear pairwise bacteriocin interaction and two on a saturated pairwise interaction[38]. We used Bayesian model selection to select the best fitting model (Supplementary Fig. S9A). The saturated model for bacteriocin interaction fitted the data best with parameters that allows for reduced growth rate and carrying capacity in the bacteriocin-producing strains (Fig. 6A, B).

The posterior parameters from the best model (Supplementary Fig. S9B) showed that *E. faecalis* exerts a competitive effect on *E. coli* so

they do have overlapping niches in this medium indicated by a negative $M_{12}$. The maximum killing, $M_{21}$, has a median value of −0.0223, −0.0242, −0.0240, for EntA, EntB, EntAB respectively, although the posteriors overlap somewhat due to low information in this experimental setup. The main difference (especially between EntA and EntB) is the $K_s$ value, which is the point of half maximum in the interaction curve. The posterior median curves for the inferred saturation function show bacteriocin killing quickly reaching a maximum in the case of EntA (Fig. 6C). This is consistent with Supplementary Figure S1B where there is no noticeable killing until a concentration of 2 µg of EntB. The dual system, EntAB, shows a behaviour inbetween the individual bacteriocins, with a slightly slower turn on than EntA but a slightly higher maximum (Fig. 5C). It was also observed that, because the saturation curves for EntA/EntAB are sharp, the effect on *E. faecalis* is approximately independent of the concentration of *E. coli*. This gives rise to a lower effective *E. faecalis* growth rate, which accounts for the qualitative difference between the EntA, EntAB and the EntB co-culture dynamics.

To test the predictive capabilities of the model, we simulated the effects of changing starting densities and seeding ratios in PM3-EntA versus *E. faecalis* co-cultures (Supplementary Fig. S10). The model is fit only to data for the 0.01 initial density and 1:1 seeding ratio (Supplementary Fig. S10A). Timecourses of all other conditions were

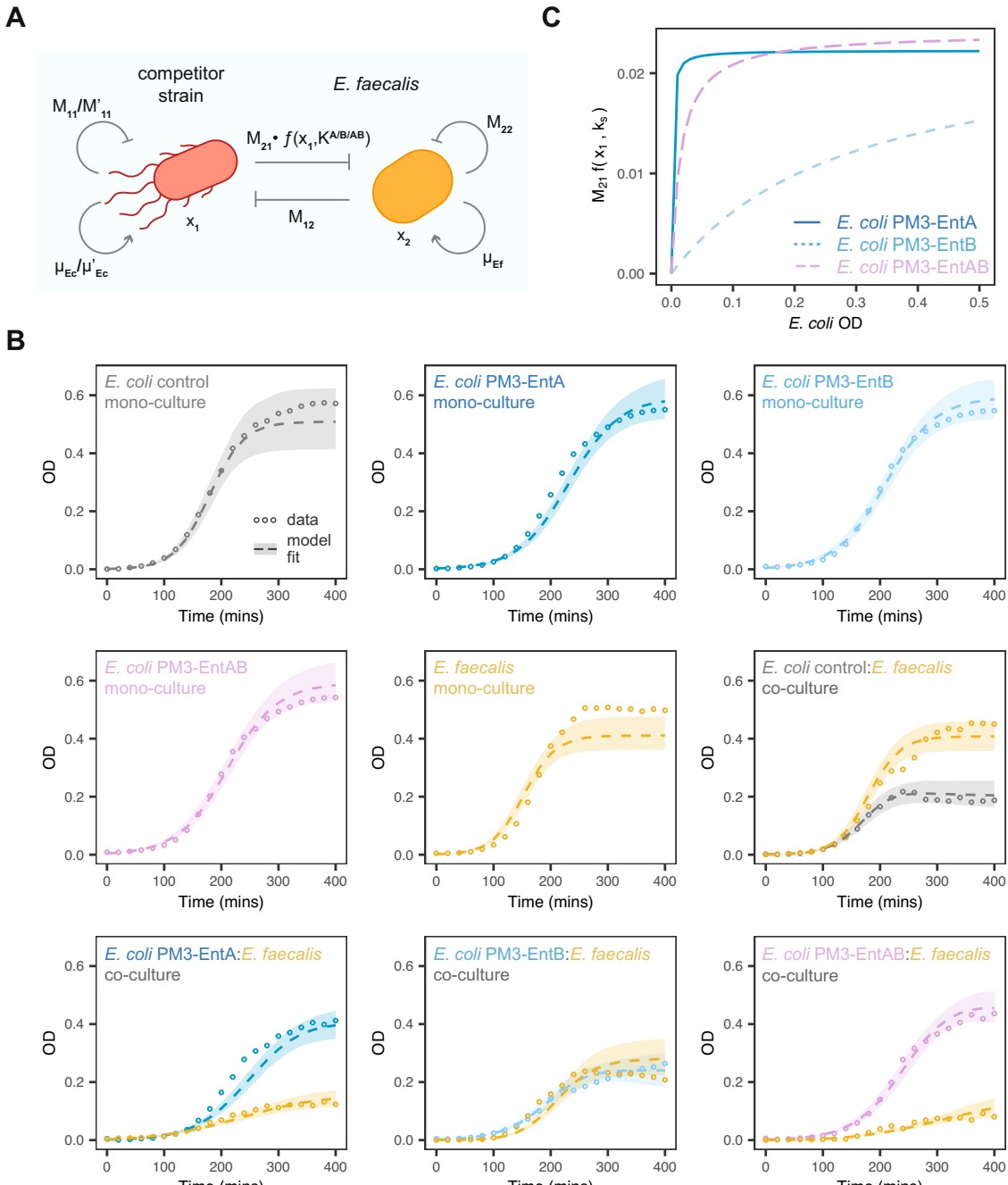

**Fig. 6 | Lotka–Volterra model fitting of co-culture growth dynamics.**
**A** Schematic of the major model parameters, $x_i$ refers to species, $\mu_i$ to species growth rate and $M_{ij}$ to species interactions. **B** Fitted growth curves for the labelled mono- and co-cultures (circles give mean experimental value, dashed lines give the median simulated timecourse and shaded regions the 95% credible region). **C** The fitted interaction term ($M_{21} \cdot f(x_1, k_s)$) against the estimated OD for the given strains, using the median of the posterior distribution. Source data are provided as a source data file.

simulated (SI Figure S10B) and used to predict the final *E. faecalis* over total OD ratio, after 6 hours. The model was able to qualitatively recapture the effects of decreasing culture density and varying seeding ratios.

## eLBP performance in a three-strain community
We further tested the activity of our bacteriocin-producing eLBPs in a three-strain community. As the third strain we used a GFP-fluorescent *E. coli* NEB®5-α strain, referred to here as the bystander species. For the control:*E. faecalis*:bystander community, *E. faecalis* was seen to outgrow the other strains in co-culture (Fig. 7B). However, in the presence

of the PM3-EntA eLBP, *E. faecalis* growth was suppressed and bystander growth dominated the co-culture. In a two-strain co-culture the *E. faecalis* strain was able to out compete the bystander strain (Fig. 7C).

## Discussion
The primary goal of this study was to create a novel AMP production platform that could be used to express bacteriocins from a commensal host. We built on previous works by developing a modular platform based on the CIDAR MoClo assembly standard[34]. Unlike previous ad hoc systems, our platform is based on a previously established cloning standard and can make use of a wide range of pre-existing parts. These

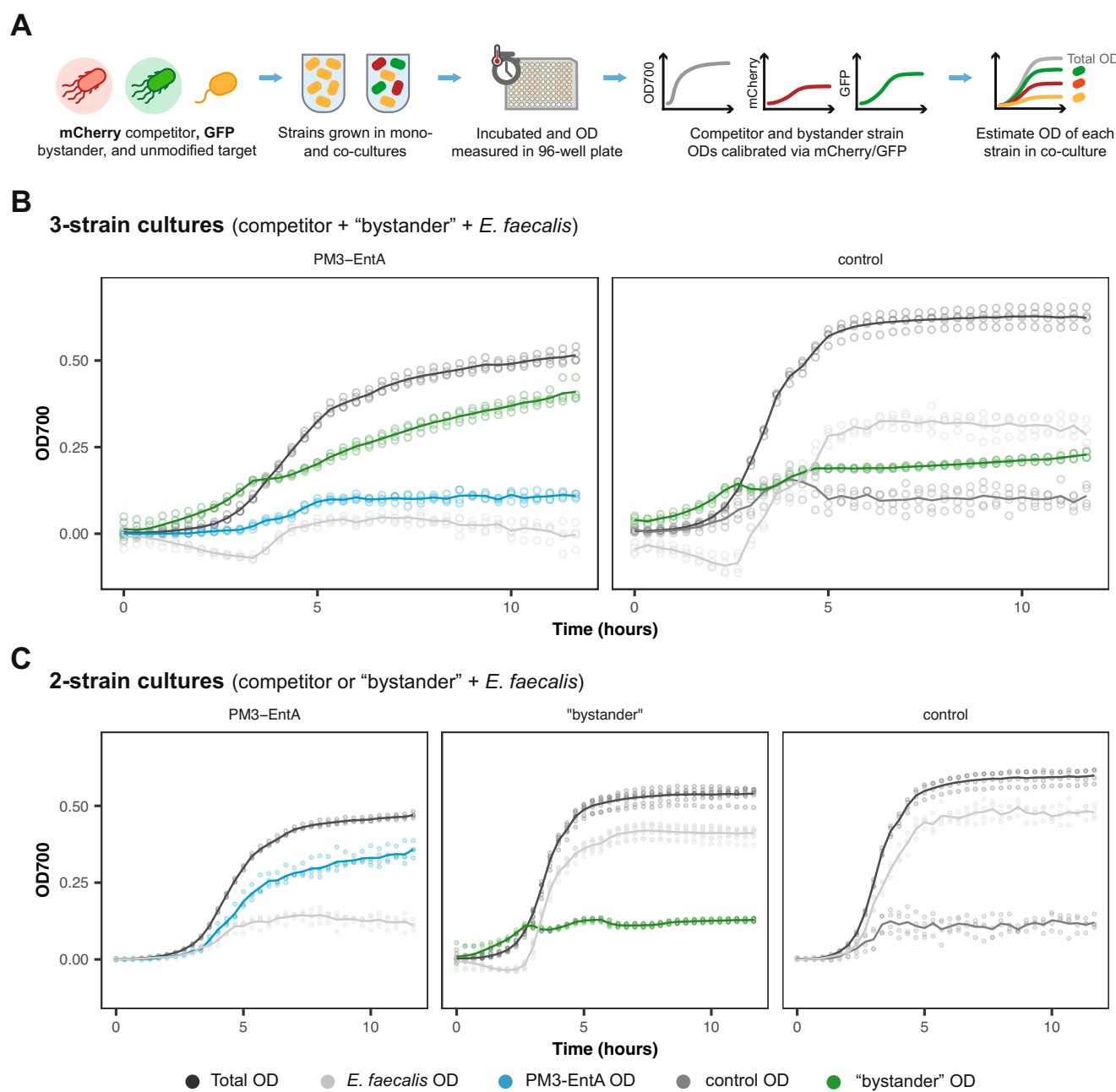

**Fig. 7 | Three-strain community culture dynamics. A** Overview of the experimental process for measuring growth in the three-strain communities. GFP and mCherry fluorescent signals are used to estimate the OD of the bystander and competitor strains, respectively. **B** Growth curves of the PM3-EntA competitor, bystander and *E. faecalis* (left panel) and the control competitor, bystander and *E.* *faecalis* (right panel) strains in co-culture. **C** Growth curves of the PM3-EntA, bystander or control strain in co-culture with the *E. faecalis* target strain in a two-strain co-culture. Lines indicate mean and points show individual measurements (*n* = 4 biological repeats). Source data are provided as a source data file.

include inducible promoters and biosensors, which allow for the tailoring of bacteriocin expression towards specific scenarios. As a demonstration, we engineered arabinose and anhydrotetracycline (aTc) inducible strains, based on the widely used ParaBAD and Ptet promoters (Supplementary Fig. S11). Our system also allows for the rapid construction of strains that are able to produce bacteriocins from separate operons. Not only does this help remove the variable of operon layout, it also allows for modification of expression levels based on interchangeable promoters and RBSs.

We screened four secretion signals: MalE, OmpA, PhoA and PM3. These tags were chosen as they are compatible with *E. coli*, have been shown to express proteins in heterologous systems, and use the general secretion (Sec) pathway[39]. The Sec pathway offers high export

capacity and a lack of specificity that is desirable for a platform that can be used to secrete a range of different bacteriocins[17]. It is worth noting that the four secretion tags explored here are generally reported as targeting protein secretion to the periplasm. A previous study suggested that proteins expressed with the PelB secretion signal are subsequently exported from the periplasm via the GspD secretion pore[40]. To confirm whether this was the case for our PM3-EntA secreting strain, a modified version of the PM3-EntA plasmid was developed including constitutive expression of the *gspD* gene (termed PM3-EntA-GspD). These constructs were then screened in both a *gspD* knockout and parent strain chassis. Even for strains with no GspD expression, inhibition zones were seen (Supplementary Fig. S12), suggesting that the GspD pore is not solely responsible for bacteriocin

secretion from these strains. This will need to be explored further, to identify which secretion pores are responsible for the activity seen with our engineered strains.

Typically the performance of AMP producing strains is quantified through supernatant growth assays that measure the growth of the target strain when exposed to either conditioned supernatant or purified AMP[23,24,41]. However, these assays do not reflect the mode of delivery in which these engineered strains are designed to operate[22]. One of the proposed benefits of using engineered bacteria is that the AMP can be expressed directly at the desired site within the body. Within this scenario the AMP secreting and target strains will likely be growing in close proximity. To more closely mimic these operating conditions we used co-culture assays to assess the performance of our bacteriocin secreting strains. We used our existing tool, FlopR, for the rapid determination of antimicrobial activity of species grown in co-culture. As the FlopR package only requires one species in a two-species co-culture to be fluorescent, it is feasible to rapidly screen antimicrobial activity against multiple strains without the need for any modification of the target species[37].

Under typical in vivo conditions other strains of the host microbiota will be present in the environment, alongside the engineered and target strains. Therefore, to further demonstrate the utility of our FlopR pipeline we characterised the PM3-EntA eLBP and control strain in a three-species community. In theory, FlopR analysis can be extended to a $n$ number of species co-culture; provided that $n-1$ distinct fluorescent signals can be measured. For the three-strain communities, GFP and mCherry signals were used to estimate the growth of the bystander and competitor strains, respectively. To achieve this we engineered a GFP expressing bystander and confirmed that other species in the co-culture did not produce a GFP signal (Supplementary Fig. S13). In a two-strain co-culture, *E. faecalis* was able to outgrow the bystander strain (Fig. 7C). This was also the case in the control:bystander:*E. faecalis* co-culture. In contrast, in the PM3-EntA:bystander:*E. faecalis* co-culture, *E. faecalis* growth was suppressed and the bystander strain was able to dominate the measured OD (Fig. 7B). This suggests that the PM3-EntA eLBP strain is able to suppress growth of the target strain, creating a niche that the bystander strain is able to exploit.

In addition, anaerobic inhibition zone assays were performed to confirm antimicrobial activity of the PM3-bacteriocin strains in anaerobic environments (such as those found within the digestive tract). All three of the bacteriocin expressing strains tested produced inhibition zones against *E. faecalis* (Supplementary Fig. S14).

EntA and EntB have been previously reported to show bactericidal activity[42]. The live/dead assay results suggested that EntA does not have bactericidal activity under the conditions tested as EntA treated cells did not have a higher percentage of dead cells than the negative control. However, the EntB treated samples had a significantly higher percentage of dead cells than the control, supporting previous reports that EntB displays bactericidal activity. Previous studies have shown that administering bacteriocins in combination can result in improved antimicrobial activity[24]. In addition, it has been reported that when co-expressed EntA and EntB can form a synergistic heterodimer[43]. The MalE-EntAB and PM3-EntAB strains suppressed *E. faecalis* growth more than the individual EntA/EntB counterparts (Fig. 4). However, OmpA-EntAB and PhoA-EntAB did not improve on the activity shown in the single bacteriocin OmpA-EntA/PhoA-EntA strains in co-culture. This may have been due to the increased metabolic burden placed on the hosts of these constructs, which is indicated by the decreased growth rate and increased lag time of these strains (Supplementary Fig. S7)[44].

Finally, we used mathematical modelling and Bayesian statistics to gain insight into the dynamics using the co-culture data. This is important because in the case of eLBPs there will be a trade-off between production of AMP and metabolic burden, with subsequent lower growth rate and reduced fitness. We tested a number of pairwise interaction models and compared how well they fit using Bayesian model selection (Fig. 6 and Supplementary Fig. S9). A model containing a saturated Lotka-Volterra term for bacteriocin interaction was the best model. Although this a common pairwise model it can also be derived from the full mechanistic model assuming that either (a) the bacteriocin-producing strain grows faster than the target or (b) that the bacteriocin is reusable[38]. Since *E. faecalis* grows faster than *E. coli* we can assume the model implies that the bacteriocin is reused after killing. We gained a number of other insights into the dynamics, including the form of the saturation functions for the different bacteriocin systems, that EntB has a much higher $K_s$ value in the saturation function than EntA, and quantification of the metabolic burden of bacteriocin expression. We anticipate that as more complex eLBPs are produced over the coming years it will be increasingly important to characterise these dynamics using mathematical models.

To date we have only screened EntAB co-expressing strains that use the same secretion signal. However, using multiplex reactions it is feasible to create strains that use a mixture of secretion signals. It is possible that other combinations may be able to alleviate negative growth impacts on the co-expressing strains while also increasing antimicrobial activity. To demonstrate these multiplex reactions, we performed a screen of PM3-EntA expressing constructs with multiple promoters and RBS parts (see Supplementary Fig. S15). From this multiplex reaction, we were able to produce eLBP strains that showed a range of antimicrobial activity. We advocate using co-cultures and mathematical modelling to assess eLBP performance. However, even in co-culture, the conditions found within the human gut will be extremely different to those encountered in vitro. Therefore, in vivo testing or improved in vitro methods will be required to more accurately predict final strain performance[45]. We also measured co-culture dynamics only indirectly using FlopR. To confirm the estimated reduction in *E. faecalis* growth shown in the FlopR assays was correct, an assay was developed in which the eight-hour timepoint of co-culture growth was sampled and used to estimate colony counts of *E. faecalis* growth when grown with the PM3-bacteriocin secreting strains. This assay confirmed that the EntA and EntAB constructs reduced *E. faecalis* growth in comparison to the control strain (see Supplementary Fig. S16). As was seen in the FlopR assays, the EntB construct had a much smaller impact on *E. faecalis* growth. Although we used pairwise mathematical models to interrogate the co-culture dynamics, mechanistic models where bacteriocin concentration is explicitly modelled would provide separate information on production and killing. Although measuring bacteriocin concentration over time in a high-throughput manner is probably infeasible, more information could be leveraged by combining co-cultures with synthetic bacteriocin spike-in experiments. Combined with Bayesian hierarchical modelling, this could allow separation of the production and killing rates.

In a wider context, several major strategies have been proposed for the production of different AMP classes. These include chemical synthesis, transgenic plants and animals, cell-free expression systems and, as used here, recombinant microbial strains[46]. This interest has been driven by the potential AMPs hold for a range of biomedical[8], agricultural[47] and food industry applications[48]. To this end, we have developed a modular bacteriocin secreting system that can be used to suppress the growth of *E. faecalis* and VRE. Although we focused on *Enterococcus* species, the modular nature of our system can make use of expanding libraries of bacteriocins[49] to design novel antimicrobials specifically targeting other bacterial species. Additionally, our experimental approach can be applied to the rapid characterisation of other antimicrobial-secreting eLBPs. In future, eLBPs like those developed here could be an important tool against the global increase in antimicrobial resistant pathogens.

## Methods

### Strains and plasmids

Construct characterisation was performed in Brain Heart Infusion (BHI) media (Oxoid Ltd, UK). All secretion signal-bacteriocin sequences were purchased as gBlock fragments (Integrated DNA Technologies, Belgium), with any internal BsaI and BbsI restriction sites removed. A list of all reagents used in this study is given in Supplementary Table 1, alongside all strains (Supplementary Table 2) and bacteriocin DNA sequences (Supplementary Table 3). All full bacteriocin secretion plasmids were constructed using DNA parts from the standard CIDAR MoClo kit[34]. Cloning was performed in commercial NEB®5-α competent *E. coli* (New England Biolabs, USA) and plasmid sequences confirmed with Sanger sequencing. Correct plasmids were then transformed into the desired competitor strains and stored as glycerol stocks at -70 °C until needed. Unless otherwise stated all characterisation experiments were performed with commercial *E. coli* NEB®express host cells. The target, *E. faecalis* DSM2570, was purchased from DSMZ-German Collection of Microorganisms and Cell Cultures GmbH.

### Solid culture characterisation

Bacteriocin secreting strains and *E. faecalis* were inoculated from glycerol stocks and grown in BHI media for ~18 h at 37 °C with shaking. About 2 μl of the *E. faecalis* culture was then diluted in 150 μl of fresh BHI media and spread on a 30 ml 1.5% BHI agar plate and allowed to dry. Cultures of the bacteriocin secreting strains were then adjusted to $OD_{700}$ 1.0 in sterile BHI media. Loading wells were stamped in the dried BHI agar plates using a sterile glass Pasteur pipette. Adjusted culture (50 μl) was added to the loading wells and the plates incubated for ~18 h at 37 °C. Assay plates were then imaged using a LoopBio Imager and inhibition zones measured manually in FIJI software[50]. The inhibition zone was defined as the area that showed no *E. faecalis* growth minus the area of the loading well.

### Liquid co-culture characterisation

Bacteriocin secreting strains were co-transformed with a constitutive mCherry2 fluorescent plasmid. All co-transformed strains and *E. faecalis* were inoculated from glycerol stocks and grown in BHI media for 18 h at 37 °C with shaking. Cultures were then adjusted to $OD_{700}$ 1.0 and diluted 100-fold in fresh BHI media. For monocultures, 120 μl of each diluted culture was added directly to the well of a 96-well plate. For co-cultures, the desired ratio of each strain was added (up to a total of 120 μl) to each well of a 96-well plate. Plates were then incubated for 24 h at 37 °C with shaking (300 rpm, 2-mm-orbital), in a Tecan Spark plate reader (Tecan, USA). Measurements for $OD_{700}$ and mCherry fluorescence (excitation: 531/20 nm, emission: 620/20 nm, gain: 120) were collected every 20 min.

The growth of each individual strain in the co-culture was then estimated using the FlopR package. A detailed description of this process is given by Fedorec et al.[37]. In brief, a calibration curve of the fluorescence in each monoculture was used to estimate the fraction of fluorescent cells in each corresponding co-culture, based on the ratio of expected vs measured fluorescence for a given OD measurement. This fraction was then used to estimate the change in species abundance across time in each co-culture. This process is summarised in Supplementary Figure S5.

Three-strain liquid culture assays were performed as described for two-strains, with the GFP fluorescent bystander strain added as a third strain to all cultures (up to a total of 120 μl). Measurements of GFP (excitation: 488/20 nm, emission: 530/20 nm, gain: 70) fluorescence were collected alongside $OD_{700}$ and mCherry readings.

### Data analysis and visualisation

All data analysis and visualisation was performed in Rstudio (R version R4.1.2). A full list of the software versions used in this study are provided in Supplementary Table 4. Further details of the materials and methods can be found in the Supporting Information.

### Reporting summary

Further information on research design is available in the Nature Portfolio Reporting Summary linked to this article.

## Data availability

Source data are provided with this paper. Plasmids are available from the corresponding author upon reasonable request. The code and data used to produce the plots in Figures 2 to 7 are available in a Zenodo repository (https://doi.org/10.5281/zenodo.11092555). Source data are provided with this paper.

## Code availability

The code used to reproduce the modelling performed in this study are available in a Zenodo repository (https://doi.org/10.5281/zenodo.11092555).

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

## Acknowledgements

The authors thank Michael J Bland and Philippe Gabant (Syngulon SA, Belgium) for kindly providing chemically synthesised EntA and EntB bacteriocin samples, and Rui Wang for assistance with the cloning of the inducible strains and multiplex reactions. This work was supported by EPSRC award EP/W004674/1 (J.W.R., L.D., C.C., Z.F.S, K.A.O, A.J.H.F., C.P.B.), EPSRC award EP/X026892/1 (S.P.N.) and MRC New Investigator Research grant MR/W025655/1 (J.A.K.M).

## Author contributions

J.W.R. and L.D. contributed equally to this work. J.W.R., L.D., J.A.K.M., A.J.H.F. and C.P.B. conceived the study. J.W.R. and L.D. performed experiments and data analysis. L.D. and S.P.N. performed peptide expression and Tricine SDS-PAGE experiments. K.A.O. performed live/dead cell assays. C.C., Z.F.S. and C.P.B. designed and implemented the

computational model and Bayesian statistics. J.W.R., L.D. and C.P.B. wrote the manuscript with input from all authors.

## Competing interests

The authors declare no competing interests.
