## [Peer Review File · Nature Communications]

REVIEWER COMMENTS

Reviewer #1 (Remarks to the Author):

The authors present an *E. coli* expression platform designed to facilitate the production and secretion of gram-positive bacteriocins, focusing on two well-known class II bacteriocins, namely enterocin A and enterocin B, originally produced by *Enterococcus faecium* strains. The initial phase of the study involves testing various secretion signals to identify the most effective ones, with performance evaluation based on the activity of recombinant *E. coli* strains in solid culture.

Subsequent experiments extend this investigation to the coexpression of both bacteriocins, providing insights into potential synergies or interactions between the two. To assess the dynamics of these processes over time, the authors transition to a liquid co-culture setup, allowing for a more comprehensive understanding of the indicator-producer dynamics in a dynamic, fluid environment. This comprehensive approach contributes valuable data to the optimization of the *E. coli* expression platform for efficient gram-positive bacteriocin production and secretion.

The study is really comprehensive and the experimental design is very careful and leaves no loose ends. The writing is very clear, and the figures and tables are very neat. In short, it is a model study of how an *in vitro* interaction between a bacteriocin-producing organism and a given indicator should be studied. Therefore, I do not see many drawbacks or elements for improvement and I think it would be ready for publication in many high impact scientific journals.

However, I believe that the novelty and the impact that this study may generate are not at the level of such a high impact journal as *Nature Communications*. This is not the first time that expression vectors and signal peptides have been used to produce and release bacteriocins from Gram+ bacteria in *E. coli*. Likewise, it is not the first time that the dynamics between a bacteriocin producer and an indicator have been studied. Moreover, as these studies have been carried out *in vitro*, using only one producer and one receptor bacterium, and with culture media and growth conditions that are optimal for the bacteria used, I do not believe that this is a true reflection of *in vivo* behaviour.

Reviewer #2 (Remarks to the Author):

This study presents a modular bacteriocin secretion platform to express bacteriocins with antimicrobial effects against *E. faecalis* and *E. faeciales*. The co-culture growth dynamics were studied using

different Lotka-Volterra models, which provide interesting insight into the interaction mechanisms. Nevertheless, some of the claims, are not sufficiently supported by the presented evidence. These claims should be relaxed or more experiments (standards for studying antimicrobial killing and resistance) are needed.

1- Any claim related to resistance should be removed, there were no experimental or modelling tests to assess resistance, only to study susceptibility (and only in terms of inhibiting growth) to the secreted bacteriocins. For example, line 99 "showed that EntAB was able to delay the development of resistance at higher concentrations" is not true.

2- The dynamics of the antimicrobial effect were studied using OD (colony-counting was used to test the validity of FlopR assay, inhibition zone assays were used but are static pictures with limited information). Nevertheless, within the manuscript, there are many mentions of killing potential or even in some cases resistance. OD counts alive, but also dead cells are contributing, and therefore it is a measurement used to assess bacteriostatic antimicrobials inhibiting growth. The standard to study the bactericidal effect are killing curves with CFUs. Therefore some claims should be removed or relaxed to represent what the experiments are actually showing. For example in the abstract: "We show that simultaneous exposure to EntA and EntB can increase the level of Enterococcus killing and delay the development of resistance". That is not completely right, the experiments show that the bacteriocins are inhibiting growth, but not necessarily killing, which cannot be demonstrated using OD. Note also that the effect is mostly delaying growth (lag phase) or decreasing exponential growth or the stationary phase, but the slope of OD is (as expected) positive or only slightly negative at the end of the stationary phase. There are only two exceptions that would like more details from the authors:

- In S1D, why this strong decrease? is it that the bacteriocins are lysing the cells? even though it seems a very strong decrease for OD.

- In two subplots in figure S5. Is this a problem due to the method used to estimate *E. faecalis* OD?

3- Whereas in the abstract it is stated that "show strong antimicrobial activity against *E. faecalis* and *E. faecium*", experiments are focused on *E. faecalis*, being the only tests for *E. faecium* in a Figure in the supplementary info (S4).

4- The parameters of the model are estimated and it is demonstrated that the model reproduces the experiments. Have the authors tried to assess (or is there any possibility) if the model is valid to predict for example the behaviour in another of the experiments not used for the parameter estimation? This will validate the model and provide a much stronger result, as alternative models with worse fitting results could be better at predicting behaviour in different conditions, and therefore at predicting the critical mechanisms.

5- When presenting the model some assumptions are posed without supporting arguments in some of the cases. It is missing for example

- Details of why to assume a standard deviation of 1 for the distribution of the initial condition. If the argument is based on the data replicates, why it was not assumed 1 for sigma, the standard deviation of xi? Note that sigma is around 0.3
- To include a reference supporting the use of LogNormal distribution for Xi
- Why Normal distribution for Ks but uniform for other parameters?

6- Some suggestions to facilitate the understanding of the modelling:

- Include the symbols of the different model parameters in Fig S8 titles. For example, where in Figure S8C are M12 and M21? Symbols are included in some subplots, such as S8B and S8E, but without using the notation in Fig A. Probably it would help also to include in the caption the model finally selected with main mechanisms (to avoid going to S8A and later to section 1.6.4)
- The definition of symbols in the supplementary material is not presented in a standard way and therefore the reader needs to go back and forth to find the meaning (it is not defined just after presenting the symbol or the equation with the symbol). This can be solved using a table of symbols. It would help also to provide the values and units for the median value of the estimated parameters.
- Somehow, the current notation may complicate the reading because M is used for the interaction (M12 and M21), but also to name the different models (M1, M2...).
- Why the model (not necessarily the parameter estimation, but at least the final simulation) is not included in the Zenodo repository? Although the model is simple and can be easily implemented, specially because the median value of the estimated parameters is not provided in the manuscript.

Reviewer #3 (Remarks to the Author):

Title: An antimicrobial peptide expression platform for targeting pathogenic bacterial species

The manuscript entitled “An antimicrobial peptide expression platform for targeting pathogenic bacterial species” intends to create a platform that can be used to express and secrete multiple bacteriocins. They produced Enterocin A and Enterocin B Escherichia coli secreting strains that show strong antimicrobial activity against Enterococcus faecalis and characterize this activity in both solid culture and liquid co-culture. The manuscript needs severe improvement and some suggestions were provided to improve the manuscript quality.

Suggestions

1. The data shown in the manuscript are about the bacteriocins enterocins A and B produced in *Escherichia coli*. Therefore, it would be more appropriate to switch the title to: "A bacteriocin expression platform for targeting pathogenic bacterial species" since there are too many different ribosomal and non-ribosomal AMPs that probably need many different strategies for expression
2. In lines 6 and 7 of the summary it is described that bacteriocins often have low in vivo stability and therefore, may not be effective when orally administered. However, the manuscript is about a new platform for the bacteriocins expression and does not address anything about antimicrobial peptide stability. Therefore, it would be better to remove this sentence since this problem is not particularly focused in this manuscript; Moreover, summary initially addresses the potential of bacteriocins against microbial resistance while the introduction initially addresses the impact of the microbiota on human health. Therefore, it is confusing for the reader which problem the work aims to solve. Please rewrite the text in a more comprehensive form;
3. In line 26 (page 2) it is described that eLBPs are bacterial strains. Is it not possible to use other expression systems such as yeast, plants, or mammalian cells? Please check and if was need please discuss;
4. It is described in line 70 (page 4) that the study targeted the bacteria *Enterococcus faecalis* and *Enterococcus faecium* as pathogens of interest. However, there is no data described in this manuscript regarding the bacterium *E. faecium*. Please check;
5. The term Gram-positive is wrong spelled, with a lowercase G. Please change it to capital G;
6. It is described in lines 98 and 99 (page 5) that EntAB was able to delay the resistance development at higher concentrations. Normally at high concentrations, the impact over resistance is higher. Authors are invited to describe how was this possible. Moreover, please add the concentrations used This information is not clear to the reader.
7. It is described in lines 153 and 154 (page 7) that *E. faecalis* was able to compete with the modified strains (PM3) after 48 hours in co-culture. It is not correct to say that the expression system created is efficient for the targeted killing of other pathogenic bacteria (lines 17 and 18, page 2) only with this information that is to broad. Please check;
8. The data presented in this manuscript may suggest that the systems developed can be bacteriostatic and not bactericidal. Could you please check and discuss if there is need
9. It is described in lines 50 to 54 (page 3) that "The development of eLBP platforms that can be used to deliver bacteriocins directly at the site of infection have the potential to overcome these challenges. A suitable secretion system is required to ensure that the expressed bacteriocin is able to exit the host cell and target pathogenic bacteria in the environment". However, the data shown in the present manuscript do not contain assays demonstrating the delivery of bacteriocins directly to the infection site Furthermore, an efficient secretion system does not guarantee that the bacteriocin reaches the target. The manuscript only describes the degradation that can occur in the production host (*E. coli*) and how secretion tags can avoid this problem. It is necessary to consider the pharmacokinetics of bacteriocins in in vivo assays, for example; Please discuss about

10. Bacteriocin expression and identification data such as SDS-PAGE and/or MALDI ToF analyses are not shown. Please provide the images for checking peptide production;

11. The caption for Figure 1 does not fully explain the figure. It is necessary to describe that the expressed bacteriocins will be secreted from the cytosol into the culture medium, as the figure shows a plasma membrane and arrows pointing the bacteriocin to the extracellular medium. Describe each element of the figure, for example, the acronyms GspD and SecYEG, and the question mark;

12. In figures 2 and 3 authors demonstrate the bacteria *E. faecalis* growth inhibition. How do you ensure that it is the bacteriocins produced that are inhibiting the bacteria's growth and not another molecule present in the culture? More precise tests using a compound purified need to be carried out to confirm this. It would be necessary to purify the bacteriocins, remove the secretion tags, and then perform antimicrobial activity assays;

13. The concept of a dual bacteriocins platform is quite interesting, but some questions are drawn at this point. First, are the bacteriocins synergistic due to different mechanisms of action or specificities? Second, another possibility is that dual activity occurs just because the dual expression strategy improves the bacteriocins concentrations and these facts lead to a higher antibacterial activity. In any case, authors are invited to add such experiments with purified and/or synthetic bacteriocins showing that at standard concentrations, a higher bactericidal activity was reached.

14. Finally and not less importantly, authors must describe a perspective of their expression strategy in a panorama of AMPs production for pharmaceutical and biotechnology industries.

Response to reviewers

Manuscript ID: NCOMMS-23-55890

We thank all three reviewers for their time and comments. The comments provided were insightful and we believe they have helped to improve the quality of our manuscript. We hope that we have answered all comments satisfactorily and that you will now consider our revised manuscript ready for publication in *Nature Communications*.

In response to reviewer's comments, we have added several new results to the manuscript:

- 3 strain co-culture of PM3 bacteriocin strains (main manuscript Figure 7). Description of additional fluorescent signal to work out species OD in these cultures (SI Figure S13).
- Live/dead assay of *E. faecalis* cells exposed to synthetic Ent A and Ent B (SI Figure S2).
- Tricine SDS-PAGE confirmation of growth inhibition at the expected protein sizes (SI Figure S4).
- Engineered strains for inducible production of PM3-EntA bacteriocin (SI Figure S11).
- Anaerobic solid culture characterisation of PM3 bacteriocin strains (SI Figure S14).
- Library generation of PM3-EntA strains, through multiplexed MoClo reactions (SI Figure S15).

Please find a point-by-point response to all comments given below (our responses are highlighted in blue). We have also attached a highlighted PDF of the revised manuscript with all changes given in blue.

Reviewer #1 (Remarks to the Author):

The authors present an *E. coli* expression platform designed to facilitate the production and secretion of gram-positive bacteriocins, focusing on two well-known class II bacteriocins, namely enterocin A and enterocin B, originally produced by *Enterococcus faecium* strains. The initial phase of the study involves testing various secretion signals to identify the most effective ones, with performance evaluation based on the activity of recombinant *E. coli* strains in solid culture.

Subsequent experiments extend this investigation to the coexpression of both bacteriocins, providing insights into potential synergies or interactions between the two. To assess the dynamics of these processes over time, the authors transition to a liquid co-culture setup, allowing for a more comprehensive understanding of the indicator-producer dynamics in a dynamic, fluid environment. This comprehensive approach contributes valuable data to the optimization of the *E. coli* expression platform for efficient gram-positive bacteriocin production and secretion.

The study is really comprehensive and the experimental design is very careful and leaves no loose ends. The writing is very clear, and the figures and tables are very

neat. In short, it is a model study of how an in vitro interaction between a bacteriocin-producing organism and a given indicator should be studied. Therefore, I do not see many drawbacks or elements for improvement and I think it would be ready for publication in many high impact scientific journals.

However, I believe that the novelty and the impact that this study may generate are not at the level of such a high impact journal as *Nature Communications*. This is not the first time that expression vectors and signal peptides have been used to produce and release bacteriocins from Gram+ bacteria in *E. coli*. Likewise, it is not the first time that the dynamics between a bacteriocin producer and an indicator have been studied. Moreover, as these studies have been carried out in vitro, using only one producer and one receptor bacterium, and with culture media and growth conditions that are optimal for the bacteria used, I do not believe that this is a true reflection of in vivo behaviour.

We thank the reviewer for their thoughtful response. We are pleased to read that the reviewer believes this study is comprehensive and well-designed.

In terms of impact, we contest the reviewer's opinion that this study is not suitable for *Nature Communications*. In our view, this study presents several novel findings. Please find some of these major points highlighted below:

1. **The modular nature of our bacteriocin producing platform.** As our platform is based on an already widely-accepted cloning standard, it is possible to readily integrate pre-existing, previously characterised, plasmid components. To further demonstrate this principle, we have added additional results to the manuscript. These have focused on two applications:
 - i. **Inducible bacteriocin expression.** We have engineered strains for arabinose and aTc-inducible bacteriocin expression. These strains were constructed using pre-existing parts, based on the widely studied ParaBAD and Ptet promoters. (SI Figure S11)
 - ii. **Multiplex reactions for construct library generation.** Our system allows for the interchange of promoter and RBS parts, these can be multiplexed for the rapid generation of libraries of bacteriocin expressing constructs. This provides a powerful tool for optimising bacteriocin expression levels and, to the best of our knowledge, is not possible with other previously reported platforms. (SI Figure S15)

Overall, we believe that use of this Golden Gate based cloning standard will allow other research groups to adopt our system more readily than previously reported expression platforms.

2. **The development of our co-culture pipeline to assess the efficacy of bacteriocin producing strains.** To the best of our knowledge, the use of fluorescent mono- and co-cultures has not been used before in this way. This pipeline reduces the time needed for characterisation and facilitates rapid, high-throughput screening of multiple engineered and target strains. To further highlight the utility of this system, we have added characterisation of the PM3- bacteriocin producing strains in 3 strain co-cultures. This shows the FlopR pipeline can be expanded to monitor growth in more complex consortia.
3. **The use of multiple different secretion tags.** As the reviewer highlights, this is not the first study to show the heterologous expression of bacteriocins from an *E. coli* host. However, previous studies such as Geldart *et al* (2018)¹ and

Gupta *et al* (2013)² have only focused on a single secretion tag (the MccV and FlgM tags, respectively). Here we describe the use of 4 secretion tags that, to the best of our knowledge, have not been used for bacteriocin production before.

Reviewer #2 (Remarks to the Author):

This study presents a modular bacteriocin secretion platform to express bacteriocins with antimicrobial effects against *E. faecalis* and *E. faecalis*. The co-culture growth dynamics were studied using different Lotka-Volterra models, which provide interesting insight into the interaction mechanisms. Nevertheless, some of the claims, are not sufficiently supported by the presented evidence. These claims should be relaxed or more experiments (standards for studying antimicrobial killing and resistance) are needed.

1- Any claim related to resistance should be removed, there were no experimental or modelling tests to assess resistance, only to study susceptibility (and only in terms of inhibiting growth) to the secreted bacteriocins. For example, line 99 "showed that EntAB was able to delay the development of resistance at higher concentrations" is not true.

We thank the reviewer for highlighting this inconsistency. We have reworded our claims to clarify that what we delay is the point at which *E. faecalis* growth begins. References to resistance have been removed.

2- The dynamics of the antimicrobial effect were studied using OD (colony-counting was used to test the validity of FlopR assay, inhibition zone assays were used but are static pictures with limited information). Nevertheless, within the manuscript, there are many mentions of killing potential or even in some cases resistance. OD counts alive, but also dead cells are contributing, and therefore it is a measurement used to assess bacteriostatic antimicrobials inhibiting growth. The standard to study the bactericidal effect are killing curves with CFUs. Therefore some claims should be removed or relaxed to represent what the experiments are actually showing. For example in the abstract: "We show that simultaneous exposure to EntA and EntB can increase the level of Enterococcus killing and delay the development of resistance". That is not completely right, the experiments show that the bacteriocins are inhibiting growth, but not necessarily killing, which cannot be demonstrated using OD. Note also that the effect is mostly delaying growth (lag phase) or decreasing exponential growth or the stationary phase, but the slope of OD is (as expected) positive or only slightly negative at the end of the stationary phase. There are only two exceptions that would like more details from the authors:

- In S1D, why this strong decrease? is it that the bacteriocins are lysing the cells? even though it seems a very strong decrease for OD.

This decline phase is also seen in the control cultures (which are not exposed to bacteriocin), suggesting that the decrease in OD is not caused by the action of the bacteriocins. Previous publications have also observed decline phases, of varying degrees, in *E. faecalis* growth (e.g., Djimeli *et al* (2016)³ and Fan *et al* (2019)⁴), therefore we suggest that the decrease is an inherent feature of *E. faecalis* growth under the conditions tested. We should also highlight that the growth curves referred to in SI Figure 1D are collected over a 48 hr period, therefore the decrease seen is

highlighted due to the extended incubation time.

- In two subplots in figure S5. Is this a problem due to the method used to estimate *E. faecalis* OD?

Apologies, can we clarify what the reviewer is referring to with this point please?

3- Whereas in the abstract it is stated that "show strong antimicrobial activity against *E. faecalis* and *E. faecium*", experiments are focused on *E. faecalis*, being the only tests for *E. faecium* in a Figure in the supplementary info (S4).

We have moved the data characterising activity against *E. faecium* to the main manuscript. This can now be seen in Figure 5B and 5C.

4- The parameters of the model are estimated and it is demonstrated that the model reproduces the experiments. Have the authors tried to assess (or is there any possibility) if the model is valid to predict for example the behaviour in another of the experiments not used for the parameter estimation? This will validate the model and provide a much stronger result, as alternative models with worse fitting results could be better at predicting behaviour in different conditions, and therefore at predicting the critical mechanisms.

We thank the reviewer for this suggestion. We have now included SI Figure S10 where we use the model, fitted using one co-culture, to simulate timecourses in other co-cultures and make a comparison to the experimental data. We have also included the posterior predictive timecourse (0.95 credible regions). You can now see that the model does a reasonable job of recapitulating the data in these other conditions, therefore providing further evidence that the model is capturing sensible biological information.

5- When presenting the model some assumptions are posed without supporting arguments in some of the cases. It is missing for example

- Details of why to assume a standard deviation of 1 for the distribution of the initial condition. If the argument is based on the data replicates, why it was not assumed 1 for sigma, the standard deviation of xi? Note that sigma is around 0.3

- To include a reference supporting the use of LogNormal distribution for Xi

- Why Normal distribution for Ks but uniform for other parameters?

We have now updated the statistical model description including more information and some references on similar approaches. We don't assume a standard deviation of 1 for the initial condition, only the prior distribution. These priors should not be set from the data themselves, as this would be using the data twice, though it is reasonable and pragmatic to set the priors at sensible order-of-magnitude values.

6- Some suggestions to facilitate the understanding of the modelling:

- Include the symbols of the different model parameters in Fig S8 titles. For example, where in Figure S8C are M12 and M21? Symbols are included in some subplots, such as S8B and S8E, but without using the notation in Fig A. Probably it would help also to include in the caption the model finally selected with main mechanisms (to avoid going to S8A and later to section 1.6.4)

This has now been fixed (now SI Figure S9). We also revised the notation to make it clearer for the reader.

- The definition of symbols in the supplementary material is not presented in a standard way and therefore the reader needs to go back and forth to find the

meaning (it is not defined just after presenting the symbol or the equation with the symbol). This can be solved using a table of symbols. It would help also to provide the values and units for the median value of the estimated parameters.

We thank the reviewer for this suggestion. We have revised the mathematical notation in the model description and have now included SI Table 2, which contains a description of the data, and SI Table 3, which contains the description of the parameters and their posterior estimates.

- Somehow, the current notation may complicate the reading because M is used for the interaction (M12 and M21), but also to name the different models (M1, M2...). We have edited the model labels to Model 1, Model 2 etc to avoid confusion.

- Why the model (not necessarily the parameter estimation, but at least the final simulation) is not included in the Zenodo repository? Although the model is simple and can be easily implemented, specially because the mean value of the estimated parameters is not provided in the manuscript.

This was an omission, we have now included all the model fitting code in an updated Zenodo repository (doi:10.5281/zenodo.11092555).

Reviewer #3 (Remarks to the Author):

Title: An antimicrobial peptide expression platform for targeting pathogenic bacterial species

The manuscript entitled "An antimicrobial peptide expression platform for targeting pathogenic bacterial species" intends to create a platform that can be used to express and secrete multiple bacteriocins. They produced Enterocin A and Enterocin B Escherichia coli secreting strains that show strong antimicrobial activity against Enterococcus faecalis and characterize this activity in both solid culture and liquid co-culture. The manuscript needs severe improvement and some suggestions were provided to improve the manuscript quality.

Suggestions

1. The data shown in the manuscript are about the bacteriocins enterocins A and B produced in Escherichia coli. Therefore, it would be more appropriate to switch the title to: "A bacteriocin expression platform for targeting pathogenic bacterial species" since there are too many different ribosomal and non-ribosomal AMPs that probably need many different strategies for expression.

We agree with the reviewer that the revised title is more appropriate. We have changed this accordingly.

2. In lines 6 and 7 of the summary it is described that bacteriocins often have low in vivo stability and therefore, may not be effective when orally administered. However, the manuscript is about a new platform for the bacteriocins expression and does not address anything about antimicrobial peptide stability. Therefore, it would be better to remove this sentence since this problem is not particularly focused in this manuscript; Moreover, summary initially addresses the potential of bacteriocins against microbial resistance while the introduction initially addresses the impact of the microbiota on human health. Therefore, it is confusing for the reader which

problem the work aims to solve. Please rewrite the text in a more comprehensive form;

We acknowledge the confusion caused by the wording of our original summary and introduction. We have now removed references to the *in vivo* stability of bacteriocins, to help clarify the major focus of this work.

3. In line 26 (page 2) it is described that eLBPs are bacterial strains. Is it not possible to use other expression systems such as yeast, plants, or mammalian cells? Please check and if was need please discuss;

The reviewer is correct that other chassis can be used to create eLBPs. However, our study focusses solely on engineered bacteria. We have reworded to clarify this point.

4. It is described in line 70 (page 4) that the study targeted the bacteria *Enterococcus faecalis* and *Enterococcus faecium* as pathogens of interest. However, there is no data described in this manuscript regarding the bacterium *E. faecium*. Please check;

The data showing bacteriocin activity against *E. faecium* was originally provided in Supplementary Figure S4. In response to this comment (and Reviewer 2 point 3), this has been moved to Figure 5 in the revised manuscript.

5. The term Gram-positive is wrong spelled, with a lowercase G. Please change it to capital G;

We have corrected this error.

6. It is described in lines 98 and 99 (page 5) that EntAB was able to delay the resistance development at higher concentrations. Normally at high concentrations, the impact over resistance is higher. Authors are invited to describe how was this possible. Moreover, please add the concentrations used This information is not clear to the reader.

The concentrations used are provided in Figure S1D, which is referred to here. We have reworded this sentence to clarify for the reader.

7. It is described in lines 153 and 154 (page 7) that *E. faecalis* was able to compete with the modified strains (PM3) after 48 hours in co-culture. It is not correct to say that the expression system created is efficient for the targeted killing of other pathogenic bacteria (lines 17 and 18, page 2) only with this information that is too broad. Please check;

A similar point was also made by Reviewer 2. We have reworded our claims to more closely reflect the data described in this manuscript.

8. The data presented in this manuscript may suggest that the systems developed can be bacteriostatic and not bactericidal. Could you please check and discuss if there is need

We have performed a live/dead cell assay, using synthetic Ent A and Ent B bacteriocins. This assay suggested that, for the conditions tested, Ent A appeared to show bacteriostatic activity, while Ent B was bactericidal. These results have been added into the supplementary information (SI Figure S2). These results are referred to on Page 5, line 93-96.

9. It is described in lines 50 to 54 (page 3) that "The development of eLBP platforms that can be used to deliver bacteriocins directly at the site of infection have the potential to overcome these challenges. A suitable secretion system is required to ensure that the expressed bacteriocin is able to exit the host cell and target pathogenic bacteria in the environment". However, the data shown in the present manuscript do not contain assays demonstrating the delivery of bacteriocins directly to the infection site. Furthermore, an efficient secretion system does not guarantee that the bacteriocin reaches the target. The manuscript only describes the degradation that can occur in the production host (*E. coli*) and how secretion tags can avoid this problem. It is necessary to consider the pharmacokinetics of bacteriocins in *in vivo* assays, for example; Please discuss about

As per suggestion 2, we have now removed references to the *in vivo* stability of bacteriocins. As highlighted by the reviewer, further *in vivo* studies would be required to conclusively prove that our platform is able to overcome the challenges of directly delivering bacteriocins to the infection site. This characterisation in more complex, *in vivo* models is signposted as future work in the manuscript. We hope the rewording of our claims has helped clarify this point.

10. Bacteriocin expression and identification data such as SDS-PAGE and/or MALDI ToF analyses are not shown. Please provide the images for checking peptide production;

We have carried out a gel activity assay (Figure S4) where concentrated ammonium sulphate precipitations of our bacteriocins produced by our eLBPs were run on a Tricine SDS-PAGE gel and then the gel overlayed on agar seeded with *E. faecium*. Zones of clearing were observed at approximately the correct sizes for our bacteriocins.

11. The caption for Figure 1 does not fully explain the figure. It is necessary to describe that the expressed bacteriocins will be secreted from the cytosol into the culture medium, as the figure shows a plasma membrane and arrows pointing the bacteriocin to the extracellular medium. Describe each element of the figure, for example, the acronyms GspD and SecYEG, and the question mark;

We have expanded the caption for this figure to explain the reported mechanisms of action.

12. In figures 2 and 3 authors demonstrate the bacteria *E. faecalis* growth inhibition. How do you ensure that it is the bacteriocins produced that are inhibiting the bacteria's growth and not another molecule present in the culture? More precise tests using a compound purified need to be carried out to confirm this. It would be necessary to purify the bacteriocins, remove the secretion tags, and then perform antimicrobial activity assays;

Please see response to point 10. In addition, we have used synthetic bacteriocins, without secretion tags, to confirm the activity of the pure bacteriocin compounds against *E. faecalis*. These results are shown in Supplementary Figure S1. We also performed the solid and liquid culture characterisation alongside control strains. These controls are the same host as the engineered strains, lacking only the bacteriocin production plasmids. These controls showed no activity against either *E. faecalis* or *E. faecium*. Taken together, these results indicate that any activity seen is not due to the production of any other molecules in the culture.

13. The concept of a dual bacteriocins platform is quite interesting, but some questions are drawn at this point. First, are the bacteriocins synergistic due to different mechanisms of action or specificities? Second, another possibility is that dual activity occurs just because the dual expression strategy improves the bacteriocins concentrations and these facts lead to a higher antibacterial activity. In any case, authors are invited to add such experiments with purified and/or synthetic bacteriocins showing that at standard concentrations, a higher bactericidal activity was reached.

We agree that the concept of combining multiple bacteriocins is a promising avenue. It should be noted that in previous literature, it has been reported that Ent A and Ent B are able to form a dimer and show synergistic behaviour (Ankaiah *et al* (2018)⁵). Within this study we use synthetic bacteriocins to show that combining Ent AB appears to be more effective at delaying *E. faecalis* growth. These results are shown in Supplementary Figure S1D. By using the same overall concentration (1, 2 or 4 µg/ml) we showed that the lag phase of *E. faecalis* exposed to Ent AB was longer than when exposed to the equivalent concentration of Ent A or Ent B in isolation.

14. Finally and not less importantly, authors must describe a perspective of their expression strategy in a panorama of AMPs production for pharmaceutical and biotechnology industries.

This manuscript focuses specifically on bacteriocins. However, we have added references to review articles that discuss the application of AMPs more generally and have highlighted other methods that have been proposed for their production. We believe that a more in-depth discussion of these topics has been covered in the review articles referenced and falls outside the scope of this article.

References

1. Geldart, K. G. *et al*. Engineered *E. coli* Nissle 1917 for the reduction of vancomycin-resistant *Enterococcus* in the intestinal tract. *Bioeng. Transl. Med.* **3**(3), (2018).
2. Gupta, S. *et al*. Genetically programmable pathogen sense and destroy. *ACS Synth. Biol.* **2**(12), (2013).
3. Djimeli, C. L. *et al*. Impact of two disinfectants on detachment of *Enterococcus faecalis* from polythene in aquatic microcosm. *RIB* **7**, (2016).
4. Fan, T_J *et al*. *Enterococcus faecalis* Gluconate Phosphotransferase System Accelerates Experimental Colitis and Bacterial Killing by Macrophages. *Infect. Immun.* **87**(7), (2019).
5. Ankaiah, D. *et al*. Cloning, overexpression, purification of bacteriocin enterocin-B and structural analysis, interaction determination of enterocin-A, B against pathogenic bacteria and human cancer cells. *Int. J. Biol. Macromol.*, (2018).

REVIEWERS' COMMENTS

Reviewer #2 (Remarks to the Author):

My critical comments have been thoroughly addressed in the revised manuscript. Text was revised to be more precise about the antimicrobial effect, new experiments were included to study the type of effect (bactericida/bacteriostatic) and the modelling approach was extended and shared zenodo. The code, including not just model simulations but calibration, was tested for some of the cases and is working perfectly fine.

Final comment in point 2 was the only one without an answer because of a problem when transcribing my notes, I apologise for this. The intention was to ask why the decrease in OD of faecalis just after starting the experiment in two subfigures in, what is now, Figure S6. This decrease is also in new figure 7.

In the model validation, only the predictions are shown but are not compared with a new set of experimental data not used for the modelling selection and estimation. This could have been a very interesting point to defend the performance of the model. In any case, the model performance seems adequate showing the expected behaviour.

Reviewer #2 (Remarks on code availability):

I have tested some of the cases (not all the different calibrations and simulations) and it was working perfectly ok.

Reviewer #3 (Remarks to the Author):

The manuscript was revised and could be accepted in the present form.

Response to reviewers

Manuscript ID: NCOMMS-23-55890B

We thank all the reviewers for their time and comments.

Please find a point-by-point response to all comments given below (our responses are highlighted in blue).

Reviewer #2 (Remarks to the Author):

My critical comments have been thoroughly addressed in the revised manuscript. Text was revised to be more precise about the antimicrobial effect, new experiments were included to study the type of effect (bactericida/bacteriostatic) and the modelling approach was extended and shared zenodo. The code, including not just model simulations but calibration, was tested for some of the cases and is working perfectly fine.

Final comment in point 2 was the only one without an answer because of a problem when transcribing my notes, I apologise for this. The intention was to ask why the decrease in OD of faecalis just after starting the experiment in two subfigures in, what is now, Figure S6. This decrease is also in new figure 7.

We believe these drops may be noise introduced during the FlopR process. As stated in the original article describing FlopR¹, the normalisation process may introduce noise at the earlier timepoints and this may account for the drops seen.

The 'negative' ODs seen at the earlier timepoints in Figure 7 are artefacts of this FlopR normalisation process, caused by noise in estimating subpopulation ODs from their respective fluorescent signals. In this case, negative ODs are not possible, but rather indicate very low populations of *E. faecalis* at the recorded timepoint.

As stated in our manuscript FlopR only allows for the estimation of subpopulation density and does not provide direct measurements.

In the model validation, only the predictions are shown but are not compared with a new set of experimental data not used for the modelling selection and estimation. This could have been a very interesting point to defend the performance of the model. In any case, the model performance seems adequate showing the expected behaviour.

Applying the fitted model to predict new experimental conditions was performed. This is shown in SI Figure S10A. Here the model parameters were estimated using only the experimental condition highlighted in red. The simulations were then performed for all other seeding ratio and dilutions values. From these results we show the fitted model is able to qualitatively recapture the effect of changing seeding ratios and

dilution values (i.e. *E. faecalis* dominates the co-culture population at low seeding densities and higher Ef:Ec ratios).

Please see lines 186-192 in the revised manuscript.

Reviewer #2 (Remarks on code availability):

I have tested some of the cases (not all the different calibrations and simulations) and it was working perfectly ok.

We are pleased that the updated code repository is functioning correctly.

Reviewer #3 (Remarks to the Author):

The manuscript was revised and could be accepted in the present form.

References

1. Fedorec, A.J.H., Robinson, C.M., Wen, K.Y., Barnes, C.P.: FlopR: An open source software package for calibration and normalization of plate reader and flow cytometry data. *ACS Synthetic Biology* 9, 2258–2266 (2020)